# Quantitative Sensory Interpretation of Rheological Parameters of a Cream Formulation

**Deborah Adefunke Adejokun and Kalliopi Dodou ***

School of Pharmacy and Pharmaceutical Sciences, Faculty of Health Sciences and Wellbeing, University of Sunderland, Sunderland SR1 3SD, UK; bg69bo@research.sunderland.ac.uk
* Correspondence: Kalliopi.dodou@sunderland.ac.uk; Tel.: +44(0)191-515-2503

**Abstract:** As the popularity of a cosmetic product on the market extensively depends on consumers' perception, it is important for the sensory evaluation to be accurate during the product developmental stage. The focus of this study was to develop a generic method for the quantitative assessment of the sensory attributes of cosmetic creams. Four 100 g oil-in-water (O/W) model creams, containing loaded niosomes and their baselines (without niosomes), were formulated. Quantitative sensory evaluation of the formulated oil-in-water products were performed in three different stages: (a) appearance—pourability (b) pick-up—firmness and elasticity/stretchability (c) rub-out—spreadability and stickiness, using rheological measurements. All measurements were carried out at skin temperature, 32 ± 1 °C, and a relative humidity (RH) of 33%. The quantitative analysis showed all cream models exhibited shear-thinning, non-Newtonian behavior. Rheological parameters from the yield stress, amplitude sweep and frequency sweep tests were found to provide realistic correlations for the sensory characteristics of pourability and spreadability, firmness, elasticity/stretchability and stickiness, respectively. This novel quantitative assessment method of the sensory characteristics of a cream proved to be highly effective and can be universally applied.

**Keywords:** Rheology; sensorial; firmness; pourability; spreadability; stickiness; stretchability

## 1. Introduction

The success of a cosmetic product on the market is largely dependent on the consumers' perception and the organoleptic profile of the product [1,2]. Therefore, sensory analysis of such a product is a mandatory process that determines market approval [2]. In 1979, sensory analysis was invented in France by Gonnet and Vache using conventional methods; these methods were later developed in Italy into a more sophisticated protocol involving extensive training, selection of panelists and sensory descriptive terms, before being absorbed by other European countries [3–7]. A general guidance for sensory analysis was then set up by the International Organization of Standardization, Geneva, ISO [8].

The sensory assessment method outlined by the ISO standard allows the qualitative and quantitative description of the attributes of a cosmetic product, hence providing accurate measurements [9]. It is a widely used tool during the product development stage, involving the adoption of a simple descriptive lexicon, a controlled environment, and 10–20 extensively trained panelists or judges that qualify the products provided based on their honest verbal perceptions (i.e., feel, fragrance and appearance), as well as quantifying the test products by assigning scores to each perception or attribute on a scale. A statistical (ANOVA) tool is then applied to compare the attributed scores and performance evaluation of the individual judges to assess data reproducibility and quality, respectively [9,10].

Sensory evaluation study performed by Gilbert et al., on eight oil-in-water cosmetic creams, using a set of panelists, successfully described perception terminology in three different stages of

simple descriptive lexicons—appearance, pick-up and rub-out—to help provide information on the identity and quality of the creams [11]. Another study performed by Montenegro et al., used the standard ISO and three-stage simple descriptive lexicon method to assess sensory attributes, however, the result showed a number of variations in the data obtained from panelists; for example, 50% of the panelists labelled three test products as oily, while the other 50% labelled the same products as non-oily [12]. The difference proved that this method is not 100% accurate due to individual preferences or limitations in sensory skills. Other limitations include the lack of adequate analytical information to back claims. It is also extensively time consuming (ranges from 10 to 120 hours based on sample nature), and expensive to acquire and maintain well-trained judges for both small and big companies and for academic research purposes, wherein time luxury and the availability of funds cannot be afforded [13]. Therefore, the need for an inexpensive, less time consuming and a more quantitative approach is essential.

The rheometer is a laboratory equipment that provides quantitative information on a product's attributes and/or qualities, by measuring flow (viscometry test, i.e., yield stress) and deformation (oscillatory test, i.e., strain/stress amplitude sweep and frequency sweep) behavior of a sample [14]. Yield stress is an important rheological parameter that allows the investigation of the critical value or amount of applied force needed to cause the structured cream to flow out of a plastic tube or be dispensed from a bottle, i.e., stress required to trigger flow. Beneath this critical value, the cream is said to deform elastically, like a solid, but flows like a liquid above the critical value [15–18].

An oscillatory rheological test that measures the degree of linearity of the formulation is the strain (stress) amplitude sweep test, a good first step in determining the viscoelastic characteristic of the cream. The linear viscoelastic region (LVR), which is the region in which a sample is capable of maintaining its structure when force is applied (the line perpendicular to the shear strain axis), gives information on cream structure/firmness, i.e., the longer the LVR, the more firm/structured the cream, while the shorter the LVR, the less firm it appears [19]. Another oscillatory rheological test is the frequency sweep test, providing structural identity, i.e., is the cream more elastic/bouncy, just like a solid or viscous like thin oil/water. The identity of the cream at a strain below the critical strain allows the assessment of the effect of colloidal forces, as well as particles and droplets interaction; the dispersed particles and/or globules are expected to float and not form sediment when the elastic (storage) modulus, G′, is greater than viscous (loss) modulus, G″, at a low frequency [20,21]. A structured or solid-like cream shows an elastic modulus or component, G′, nearly independent of frequency, while the more dependent G′ is on frequency, the more liquid the cream. The cream is said to be non-sticky when there is no crossover of the elastic G′ and viscous G″ moduli, and sticky in nature when a crossover occurs [21].

The association between rheological measurements and the adhesive ability (tackiness) of pressure-sensitive adhesives on the skin is well-known [19] and evidences that there is a correlation between user trial data with rheological measurements. The objective of this study was to expand this association to a wider range of sensorial attributes by developing a standard, simple and reliable method for the quantitative assessment of the sensorial attributes of O/W cream formulations by correlating simple sensory lexicons to viscometry (yield stress) and oscillatory (amplitude and frequency sweep) rheological parameters.

## 2. Materials and Methods

### 2.1. Materials

The active ingredient (X), cholesterol, span65 and solutol HS-15 were obtained from Sigma-Aldrich, Inc. (Gillingham, UK). Baobab oil was purchased from Aromatic Natural Skin Care (Forres, UK), Jojoba and Coconut oil from SouthernCross Botanicals (Knockrow, Australia). The Emulsifying Wax was obtained from CRODA International Plc (Goole, East Yorkshire, UK). Other excipients of the cream and Tris buffer solutions were of analytical grade.

## 2.2. Methods

### 2.2.1. Preparation of Niosomes

Five (5) niosome formulations, labelled A to E, were prepared using the thin-film hydration technique, with cholesterol (45%), span 65 (45%), solutol-HS 15 (10%) each dissolved in 4 mL organic solvent (chloroform) in a 250 mL round bottom flask, for the manufacture of 300 μmol of vesicles. The chloroform was removed using a rotary evaporator at 60 °C, 40 rpm and a vacuum of 464 ± 10 mbar. After placing the 250 mL flask at the interface of the $H_2O$ in the bath, the pressure was allowed to drop until no chloroform was left and a thin film of the mix formed on the flask wall (Figure 1). A total of 5 mL of Tris buffer pH 7.4 with 0.01 mL or 10 uL of the active, X (i.e., total active concentration added was 0.002 v/v) was added to hydrate the lipid films, followed by gentle agitation—enabling the formation of multi-lamellar vesicles and the entrapment of the active in the vesicles. The mix was intermittently incubated at 60 °C for a period of 10 minutes while shaking to allow for the complete detachment of the lipid film, encouraging more entrapment. After this, the newly prepared niosomes were separated via sephadex G-50 column chromatography and characterized. According to results obtained from characterization studies on all five niosomal formulations, models C and D were proven to be of excellent quality (i.e., sizes of 592 and 601 nm and −49.2 and −34.5 mV surface charge, respectively) and were therefore considered lead formulations and incorporated into the cream base via manual mixing.

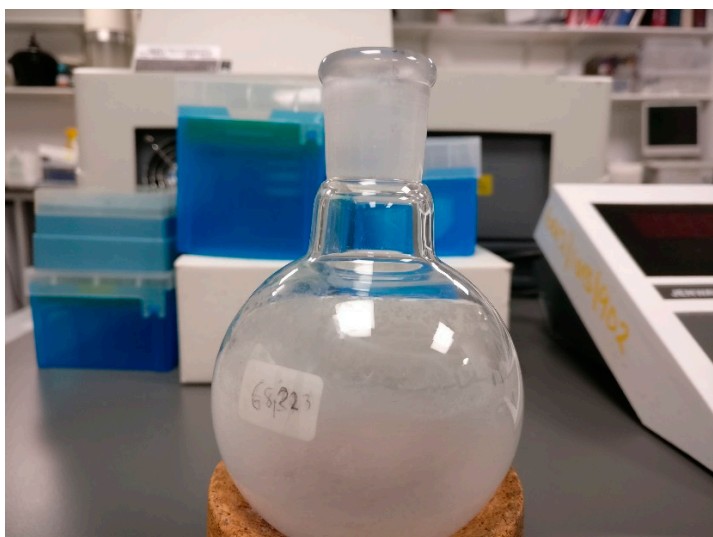

**Figure 1.** A photographic image of the thin film formed around the flask wall.

### 2.2.2. Preparation of Creams

Four (4) 100 g O/W model creams containing active-loaded niosomes (labelled as model IA-IVA), and their baselines without niosomes (labelled as model IB-IVB), were prepared with the formulas stated in Table 1, according to the following method: the oil phase and water phase ingredients were weighed in two separate beakers. After heating the oil phase and water phase to 75 °C, both phases were mixed together for 18 minutes at 9500 rpm using the Silverson L5M electric homogenizer to obtain a uniform mix. At a cool down temperature of 40 °C, 5% of the active niosomes suspended in water were added to each cream model in batch "A" and further mixed manually with a glass stirrer for two minutes, to avoid disruption of the vesicles. The newly formulated products were collected into eight (8) separate 100 g glass jars with plastic caps, labelled IA-IVA and IB-IVB, with and without actives, respectively. The first, labelled model I (1:1 of jojoba and baobab oil) contained a water phase of 85%, oil phase (10%) and emulsifier (5%) while the remaining three had an equal % composition of

water phase (83%), oil phase (12%) and emulsifier (5%), labelled II (1:1 of jojoba and baobab oil), III (1:1 of jojoba and coconut oil) and IV (1:1 of baobab and coconut oil).

**Table 1.** Ingredient and amount variables in 100 g of each cream formulation.

| Phase | INCI | Composition (%) | IA | IB | IIA | IIB | IIIA | IIIB | IVA | IVB |
|---|---|---|---|---|---|---|---|---|---|---|
| | Stearyl Alcohol | Stearyl Alcohol | | | | | 1 | 1 | 1 | 1 |
| Oil | Simmondsia Chinensis Seed Oil | Jojoba Oil | 4 | 4 | 5 | 5 | 5 | 5 | | |
| | Adansonia Digitata Seed Oil | Baobab Oil | 4 | 4 | 5 | 5 | | | 5 | 5 |
| | Cocos nucifera | Coconut Oil | | | | | 5 | 5 | 5 | 5 |
| | Glycerin | Glycerine | 5 | 5 | | | 5 | 5 | 5 | 5 |
| Water | Propylene Glycol | Propylene Glycol | | | 5 | 5 | | | | |
| | Aqua | Water | 73.7 | 78.7 | 71.7 | 76.7 | 71.7 | 76.7 | 71.7 | 76.7 |
| Active | - | Entrapped Active | 5 | | 5 | | 5 | | 5 | |

This study was a part of a wider study involving the design of a novel cream containing active ingredient (X), which has not yet been exploited, for the treatment of an aesthetic condition. Therefore, the authors do not wish to disclose the identity of the active contained in the niosomes.

### 2.2.3. Sensory Lexicons and Definitions

A sensory lexicon was devised in three different stages [11,12], for all formulated oil-in-water products: (a) appearance—pourability (b) pick-up—firmness and elasticity/stretchability (c) rub-out—spreadability and stickiness. Each stage was correlated with rheological parameters, as shown in Table 2, to help provide information on the identity and quality of the test products.

**Table 2.** Proposed protocol of rheological parameters–sensory attribute pairs, and their description.

| Stage of Usage | Sensorial Attribute | Description | Rheological Parameter |
|---|---|---|---|
| Appearance | Pourability | Ability of a product to flow or be pumped out of the container when a force is applied. | Viscometry; Yield Stress |
| Pick-up | Firmness | The degree to which the product is able to hold its shape or structure in the presence of force. | Oscillatory; Amplitude Sweep |
| | Elasticity/ Stretchability | It is the ability of the product to deform or expand (strain) by resisting an external force (stress). | Oscillatory; Frequency Sweep |
| Rub-out | Spreadability | The force required to cause flow of the product. | Viscometry; Yield Stress |
| | Stickiness | Ability of product to attach to the skin, yielding a sticky skin feel. | Oscillatory; Frequency Sweep |

### 2.2.4. Instrumental Rheology and Sensory Characterization

To obtain the rheological measurements of the cream models, a Kinexus lab+ Rotational Rheometer (Malvern Panalytical Instruments, Malvern, UK) was used with a stainless steel parallel plate of 20 mm

diameter at a constant temperature of 32 ± 1 °C, a gap size of 0.25 mm, and a humidity of 33%. All measurements were performed in triplicate (n = 3)

1. Yield Stress: pourability and spreadability—a stress range of 0.001 Pa to 10,000 Pa at a ramp time of 2 min and a decade of 10 was applied.
2. Strain Amplitude Sweep with LVR Determination: firmness—the samples were oscillated over a shear stress range of 0.001 Pa to 10,000 Pa, at a frequency of 1 Hz and a decade of 10.
3. Frequency Sweep: stickiness and elasticity or stretchability—the samples were oscillated over a frequency range of 50 to 0.05 Hz, at a % strain within the LVR.

### 2.2.5. Statistical Analysis

Statistical evaluation of results obtained for the formulated creams was achieved using the SPSS software (SPSS UK Ltd, IBM, Woking, UK). To indicate whether any significant correlations ($p <$ 0.05) exist between the rheological data obtained on all eight O/W creams, Pearson's Chi-square test was conducted.

## 3. Results and Discussion

### 3.1. Rheology and Sensory Characterization

#### 3.1.1. Yield Stress: Pourability and Spreadability

This is an important parameter as it allows the investigation of the amount or critical value of applied force needed to cause the structured cream to flow out of a plastic tube or be dispensed from a bottle, i.e., the stress required to trigger pumping through a pipeline. Beneath this critical value, the cream is said to deform elastically like a solid, but it flows like a liquid above the critical value [15,16]. Therefore, two types of information on the pourability (yield stress value) and spreadability (viscosity value) of the measured product, where 0 signified the least pourable or spreadable score and 9 indicated the most pourable or spreadable score, are reported in Table 3. The scale ranges of 0–9 (Table 3) and 0–3 (Table 4) were carefully selected to provide distinct groups of similar values that would be statistically significant from each other.

**Table 3.** Correlation of the range of yield stress, viscosity values and amplitude sweep to pourability, spreadability and firmness scores (0–9).

| Score | Yield Stress Values (Pa) | Viscosity/Thickness (Pa S) | Strain Amplitude Sweep (Pa) |
|---|---|---|---|
| 0 | 181–200 | 171,000–190,000 | <0.010 |
| 1 | 161–180 | 151,000–170,000 | 0.011–0.020 |
| 2 | 141–160 | 131,000–150,000 | 0.021–0.040 |
| 3 | 121–140 | 111,000–130,000 | 0.041–0.060 |
| 4 | 101–120 | 91,000–110,000 | 0.061–0.080 |
| 5 | 81–100 | 71,000–90,000 | 0.081–0.100 |
| 6 | 61–80 | 51,000–70,000 | 0.101–0.200 |
| 7 | 41–60 | 31,000–50,000 | 0.201–0.400 |
| 8 | 21–40 | 11,000–30,000 | 0.401–0.600 |
| 9 | 0–20 | <10,000 | 0.601–0.800 |

**Table 4.** Correlation of the range of yield stress, viscosity values and amplitude sweep to pourability, spreadability and firmness scores (0–3).

| Score | Yield Stress Values (Pa)—Pourability | Viscosity/Thickness (Pa S)—Spreadability | Strain Amplitude Sweep (Pa)—Firmness |
|---|---|---|---|
| 0 | 151–200 | 151,000–200,000 | <0.200 |
| 1 | 101–150 | 101,000–150,000 | 0.201–0.400 |
| 2 | 51–100 | 51,000–100,000 | 0.401–0.600 |
| 3 | 0–50 | <50,000 | 0.601–0.800 |

Figure 2 reveals that all cream models exhibited non-Newtonian behavior, shear-thinning with increasing stress or applied force. In Table 5, model IIA was the most structurally robust, with the highest yield stress of $112/\pm22.3$ Pa (i.e., model IIA requires a large amount of force to break its structure apart, allowing it to flow like a liquid) and the highest viscosity/thickness of $117302/\pm36498$ PaS, therefore having the lowest pourability and spreadability scores of four and three, respectively. Model IIB had a low yield stress of $48/\pm15.2$ Pa and a viscosity/thickness of $34358/\pm9249$ PaS; this means that model IIB showed an increased spreadability and pourability score of seven, compared to model IIA. Model IA was the second most structurally robust, with a high yield stress of $79/\pm15.8$ Pa, indicating that a large amount of force is needed to break its structure apart, with a viscosity/thickness of $53270/\pm3010$ PaS, consequently possessing a pourability and spreadability score of six. Model IB had the lowest yield stress of $26/\pm15.5$ Pa and the highest pourability score of eight compared to other creams, i.e., it requires the least force to break its structure apart, and a viscosity/thickness of $21590/\pm10090$ PaS, with a high spreadability score of eight.

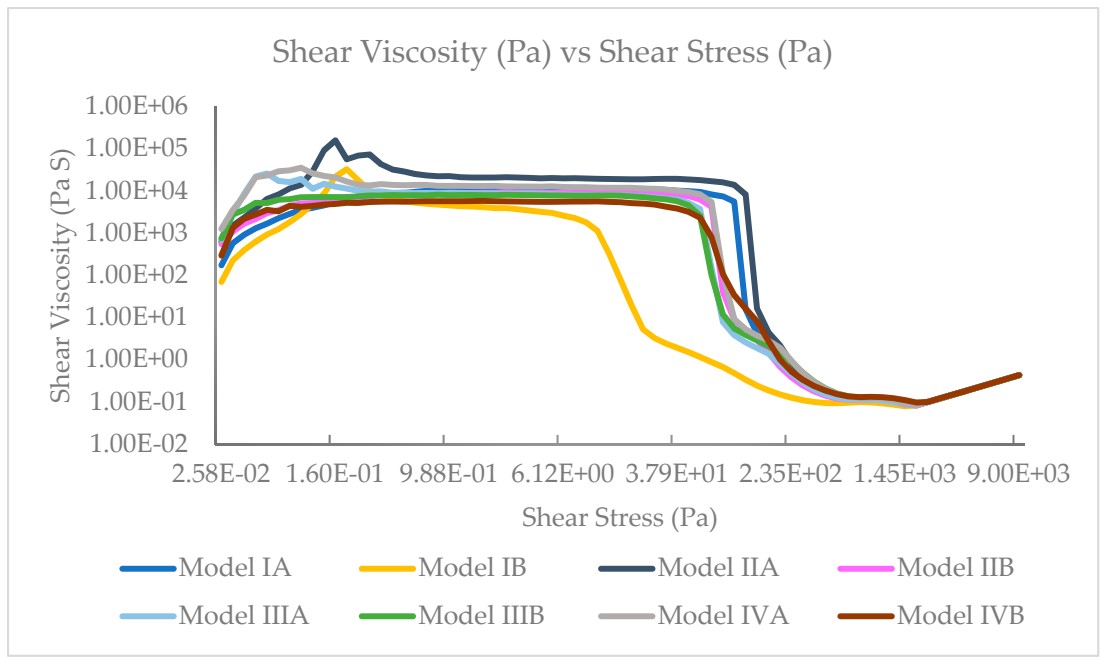

**Figure 2.** Viscosity (Pa S) of sample against applied stress (Pa).

**Table 5.** Mean and standard deviation of yield stress and viscosity/thickness values. (n=3).

| Model | Mean Yield Stress (Pa) | Pourability Score | Mean Viscosity/ Thickness (PaS) | Spreadability Score | Mean Shear Strain within the LVR | Firmness Score |
|---|---|---|---|---|---|---|
| IA | 79 ± 15.8 | 6 | 53,270 ± 3010 | 6 | 9.755E−002 ± 6.028E−003 | 5 |
| IB | 26 ± 15.5 | 8 | 21,590 ± 1090 | 8 | 7.206E−002 ± 6.513E−003 | 4 |
| IIA | 112 ± 22.3 | 4 | 117,302 ± 36,498 | 3 | 7.268E−002 ± 6.628E−003 | 4 |
| IIB | 48 ± 15.2 | 7 | 34,358 ± 9249 | 7 | 5.077E−002 ± 2.341E−002 | 3 |
| IIIA | 66 ± 10.5 | 6 | 20,100 ± 3874 | 8 | 1.022E−001 ± 9.295E−003 | 6 |
| IIIB | 67 ± 10.5 | 6 | 8085 ± 15 | 9 | 6.844E−002 ± 2.498E−002 | 4 |
| IVA | 75 ± 7.5 | 6 | 38,050 ± 4550 | 7 | 4.910E−002 ± 1.007E−002 | 3 |
| IVB | 46 ± 9.2 | 7 | 4767 ± 1067 | 9 | 1.272E−001 ± 1.905E−003 | 6 |

Model IVA also had a yield stress of 75/±7.5 Pa, with a viscosity/thickness of 38050/±4550 PaS, and a decreased pourability and spreadability score of six and seven, respectively when compared to its pair. Model IVB had a low yield stress of 46/±9.2 Pa (pourability score of seven), and the lowest viscosity/thickness of 4767/±1067 PaS, providing the highest spreadability score of nine. Model IIIB had a yield stress of 67/±10.5 Pa (pourability score of six), with the second lowest viscosity/thickness of 8085/±15 PaS compared to others, showing an increased spreadability score of nine, while model IIIA had a similar yield stress of 66/±10.5 Pa (pourability score of six) but a higher viscosity/thickness of 20100/±3874 PaS (spreadability score of eight), when compared to its pair. Models IB and IVB appeared to be the best creams in terms of pourability and spreadability scores, with eight, seven and nine, respectively. The largest difference is seen between the following pairs: IA and IB, IIA and IIB, and IVA and IVB, and could be a result of the presence of active niosomes contained in the former (IA, IIA and IVA).

Generally, it was observed that all model creams without niosomes exhibited higher pourability and spreadability scores, with a lower firmness score, when compared to their noisome-containing counterparts. This shows the sensitivity of the method in detecting the effect of niosome vesicles on the overall sensorial perception of the creams in terms of pourability, spreadability and firmness.

Pearson Chi-Square test showed a statistical correlation between viscosity and yield stress values for all eight samples with *p* values < 0.001.

3.1.2. Stress (Strain) Amplitude Sweep with LVR Determination: Firmness

An oscillatory test that measures the degree of linearity of the formulation is the strain or stress amplitude sweep test, a good first step in determining the viscoelastic characteristics of the cream. As shown on Figure 3, the linear viscoelastic region, LVR, gives information on how stable/firm/structured the cream is, i.e., the longer the LVR, the more structured the cream, while the shorter the LVR, the less structured it is. Other information, such as the position of the LVR, illustrates how well the cream is able to resist stress [19]. As reported in Table 3, score 0 signifies the least firm/structured cream while the most firm/structured product was allocated score 9.

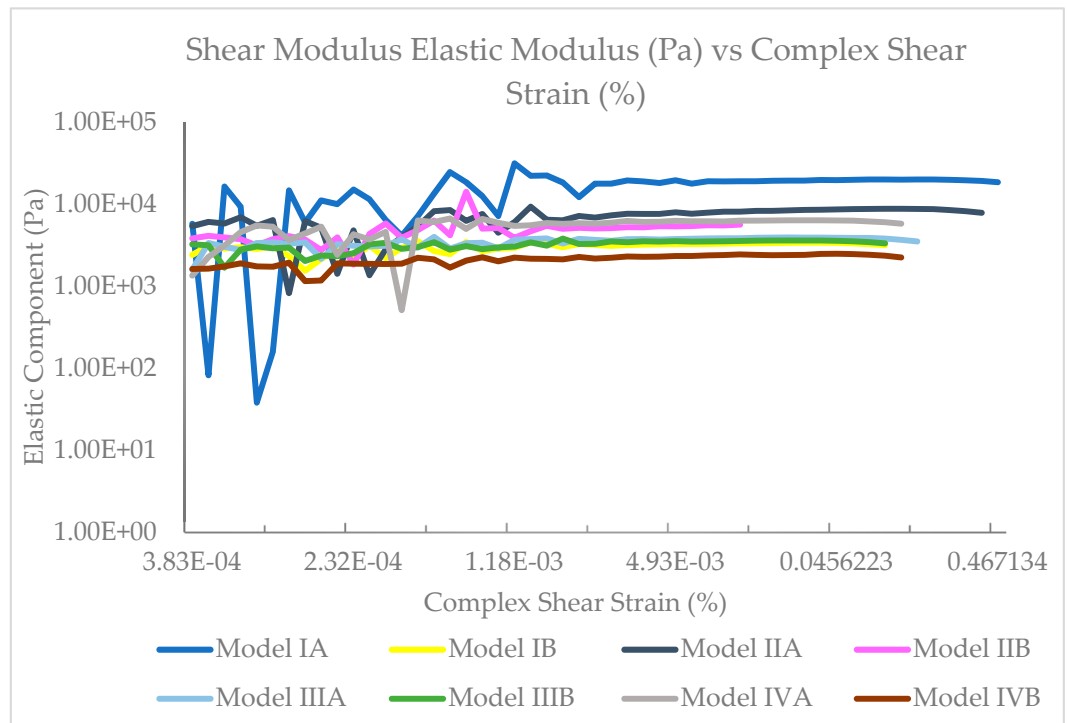

**Figure 3.** Elastic modulus, G' (Pa) plotted against complex strain (%).

Table 5 above showed the highest firmness was observed in creams containing coconut oil: model IVB (coconut and baobab oil) and IIIA (jojoba and coconut oil)—0.13 and 0.10, respectively. The least firmness was seen in model IVA and IIB—0.049 and 0.051, respectively. Therefore, the method was sensitive enough to differentiate between the effect of oils on the texture of the formulation. These results complied with the frequency data, showing the two models to be more elastic compared to models IA, IB, IIA, IIB IIIB and IVA, because IVB and IIIA are more firm and are capable of resisting the action of any external force longer than others, maintaining their structure.

3.1.3. Frequency Sweep; Stickiness and Elasticity or Stretchability

The frequency sweep test is also an oscillatory rheology test that gives information on the structure (elastic/bouncy, just like a solid, or viscous like thin oils or water) or identity of the cream at a strain below the critical strain. Therefore, allowing for the assessment of the effect of colloidal forces as well as particles and droplets interaction, the dispersed particles and/or globules are expected to float and not form sediment when G' is greater than the viscous modulus, G", at a low frequency, and vice-versa. A structured or solid-like cream shows an elastic modulus or component, with G' nearly independent of frequency, while the more dependent G' is on frequency, the more liquid the cream. The cream is said to be non-sticky when no crossover of the elastic and viscous modulus is observed, and sticky in nature when crossover occurs [21]. Score 0 represented a non-sticky or non-stretchy, while score 3 indicated a very sticky or very stretchy cream (Table 6).

**Table 6.** Correlation of Frequency Sweep Information to Stickiness and Elasticity/Stretchability Scores (0–3).

| Score | Elasticity/Stretchability | Stickiness |
|-------|---------------------------|------------|
| 0 | Non-Stretchy | Non-Sticky |
| 1 | Moderately Stretchy | Moderately Sticky |
| 2 | Stretchy | Sticky |
| 3 | Very Stretchy | Very Sticky |

The graphs in Figure S1 show that all cream models were non-sticky in nature (i.e., no crossover was observed) and had their G′ component greater than G″ at a low frequency, indicating the stability of products, as all particles and globules did not sediment or separate. However, it was observed that models IB and IIB had their G′ component a lot higher than G″ at a low frequency compared to the pairs containing active niosome particles, models IA and IIA, whereas the opposite was seen when models IIIB and IVB were compared with IIIA and IVA. (See Supplementary Materials)

It was also observed that models IIIA IIIB, IVA and IVB showed their G′ to be more independent of frequency than models IA, IB, IIA and IIB, implying that models IIIA IIIB, IVA and IVB are more solid, therefore, more elastic or stretchy in nature, and had the highest elasticity score of 2 (Table 7) compared to models IA, IB, IIA and IIB. Models IVB and IIIA also exhibited the highest firmness scores, showing similarities with the elasticity data. This could be the effect of the coconut oil contained in models IIIA IIIB, IVA and IVB.

**Table 7.** Stickiness and Elasticity or Stretchability Scores for the eight O/W Creams.

| Model | Stickiness | Score | Elasticity/Stretchability | Score |
|-------|-----------|-------|---------------------------|-------|
| IA | Non-Sticky | 0 | Moderately Stretchy | 1 |
| IB | Non-Sticky | 0 | Moderately Stretchy | 1 |
| IIA | Non-Sticky | 0 | Moderately Stretchy | 1 |
| IIB | Non-Sticky | 0 | Moderately Stretchy | 1 |
| IIIA | Non-Sticky | 0 | Stretchy | 2 |
| IIIB | Non-Sticky | 0 | Stretchy | 2 |
| IVA | Non-Sticky | 0 | Stretchy | 2 |
| IVB | Non-Sticky | 0 | Stretchy | 2 |

The sensorial properties of these formulations can be depicted in radar diagrams, as shown in Figures 4 and 5. These diagrams compare the sensorial properties of products and can be used as a marketing tool.

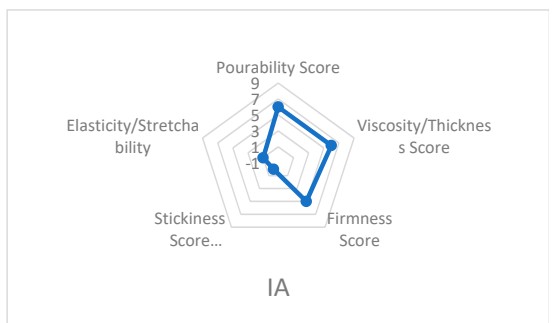

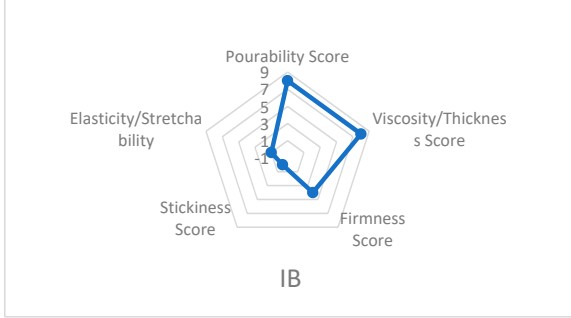

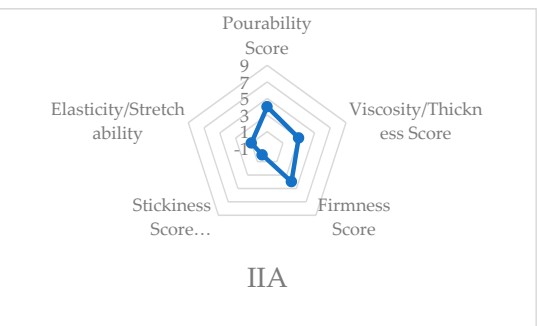

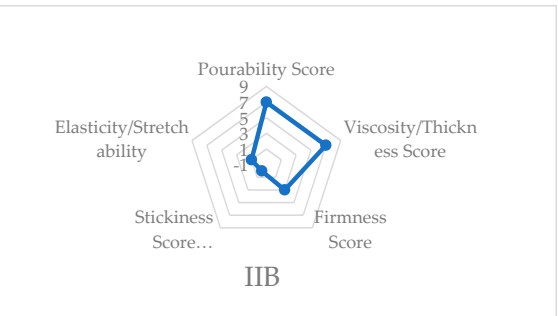

**Figure 4.** *Cont.*

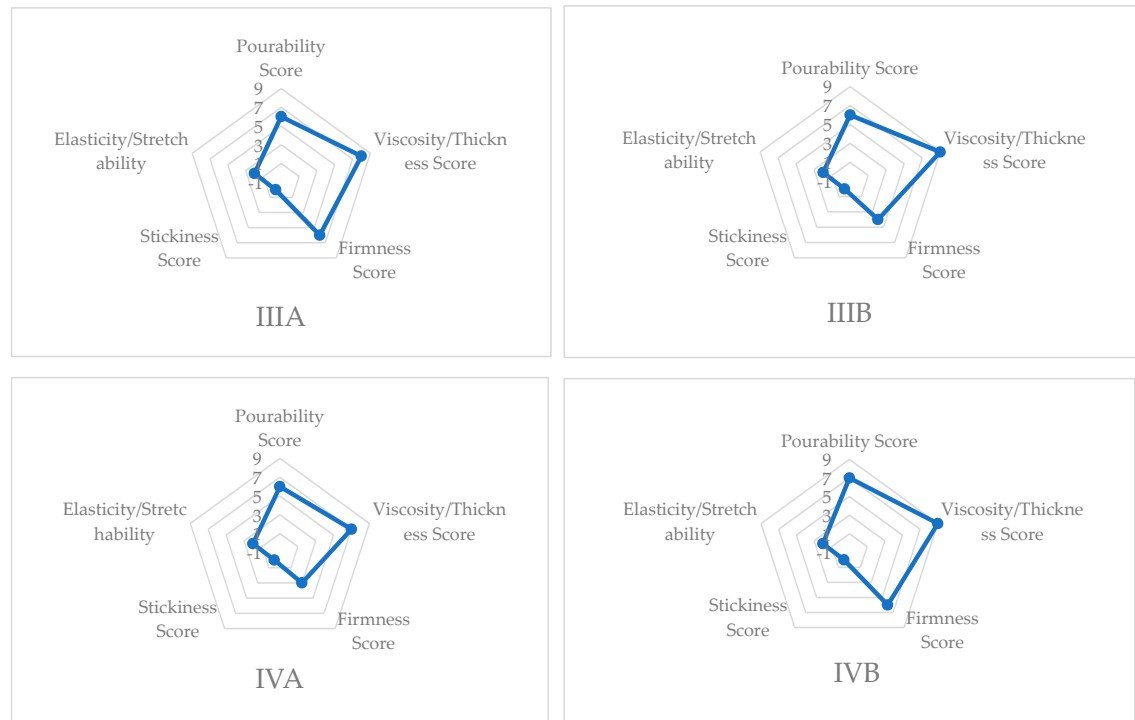

**Figure 4.** Radar diagrams of all eight oil-in-water cream model pairs (IA/IB, IIA/IIB, IIIA/IIIB, IVA/IVB) indicating Pourability, Spreadability, Firmness, Stickiness and Elasticity or Stretchability on a scale (0–9).

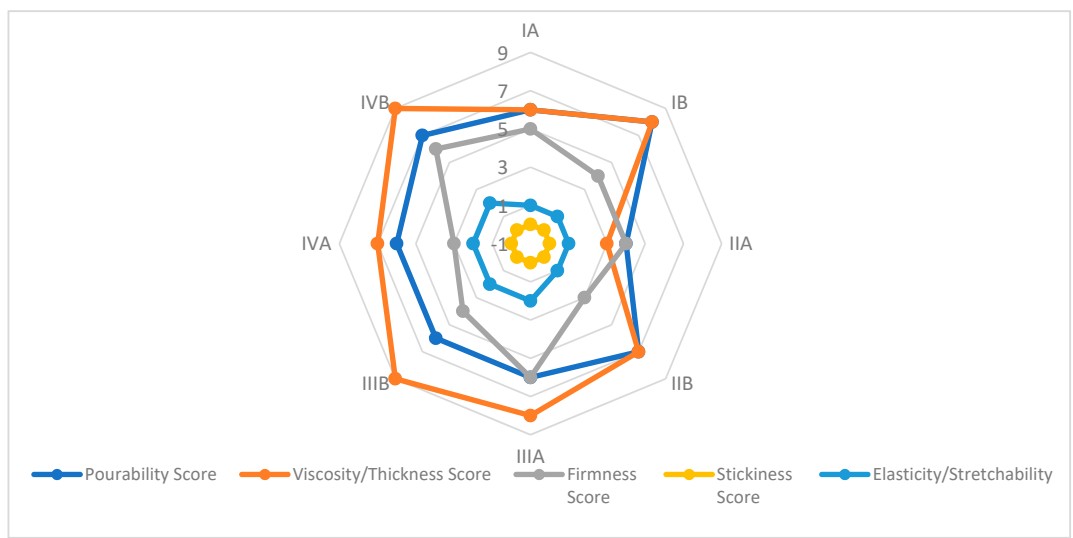

**Figure 5.** Radar diagram of the summary of all eight O/W creams indicating Pourability, Spreadability, Firmness, Stickiness and Elasticity or Stretchability on a scale (0–9).

## 4. Conclusions

As consumer perception of a cosmetic product is an important determinant of market approval and success, sensory assessment is consequently a mandatory step in the claims substantiation stage of a product's launch to the market.

In this study, we report a new test protocol, which correlates the rheological parameters of semisolid formulations (creams) with their sensorial characteristics such as pourability, firmness, elasticity and stickiness. This protocol avoids the time, costs and subjectivity associated with qualitative user-trials;

it is a quantitative method that can be used for the creation of sensorial radar diagrams for cosmetic and personal care semisolid formulations.

One limitation of the protocol is the inability of rheological measurements to reveal sensory attributes like odour, color, glossiness and oiliness. This limitation can be compensated by using other relevant analytical laboratory meters in conjunction with the rheological measurements.

**Supplementary Materials:** The following are available online at http://www.mdpi.com/2079-9284/7/1/2/s1, Figure S1: The graphs above illustrates elastic shear modulus, G′ (Pa) plotted against frequency sweep (Hz).

**Author Contributions:** Conceptualization, D.A.A. and K.D.; Data curation, D.A.A.; Formal analysis, D.A.A. and K.D.; Investigation, D.A.A.; Methodology, D.A.A. and K.D.; Project administration, K.D.; Resources, K.D.; Supervision, K.D.; Validation, K.D.; Visualization, D.A.A. and K.D.; Writing – original draft, D.A.A.; Writing – review & editing, K.D. All authors have read and agreed to the published version of the manuscript.

**Funding:** This research received no external funding.

**Acknowledgments:** The researcher would like to thank the formulation laboratory for supplying some of the materials used in the study.

**Conflicts of Interest:** The authors declare no conflict of interest.

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
