# Peer review of "Quantitative Sensory Interpretation of Rheological Parameters of a Cream Formulation"

_cosmetics, doi:10.3390/cosmetics7010002_

Round 1

Reviewer 1 Report

References :

Potential adds : [3] doi : 10.5772/51846 ; please change [14] and [18] links are not available. In addition a lot of reviews are available for example https://doi.org/10.1122/1.5085363 and https://doi.org/10.1016/j.progpolymsci.2016.12.007  

A similar method is already presented for stretching properties for cosmetic emulsions. See https://doi.org/10.1016/j.carbpol.2012.12.028

comments :

The study is relevant and requires a sophisticated understanding of the links between cognitive activity (sensitivity) and corresponding physical characteristics. it is important to explain the reason for the choice of creams presented, especially with regard to the scale from 0 to 9 deduced. Can you explain why you cover the largest panel of creams for a relevant consumer perception  ? An extended study including correlation for these products human panel study is necessary to really talk about quantitative sensory interpretation. 

Author Response

Potential adds : [3] doi : 10.5772/51846 ; please change [14] and [18] links are not available. In addition a lot of reviews are available for example https://doi.org/10.1122/1.5085363 and https://doi.org/10.1016/j.progpolymsci.2016.12.007  

We have amended the References by adding these suggested papers and sorting out the links. 

A similar method is already presented for stretching properties for cosmetic emulsions. See https://doi.org/10.1016/j.carbpol.2012.12.028

This study illustrated just a single sensory attribute (stretchability) of 9 O/W emulsions, and also failed to interpret data on a simple scaling range for easy understanding and quantitative evaluation.

Whereas in our study we describe all important human sensory attributes for any cream formulation such as: pourability and spreadability; firmness; elasticity/stretchability and stickiness, that would allow their schematic quantitative representation on a radar diagram.  

comments :

The study is relevant and requires a sophisticated understanding of the links between cognitive activity (sensitivity) and corresponding physical characteristics. it is important to explain the reason for the choice of creams presented, especially with regard to the scale from 0 to 9 deduced. Can you explain why you cover the largest panel of creams for a relevant consumer perception  ?

We have added a few sentences on the paper to explain the background rationale for the choice of creams (p.4) and how the scaling system was deduced (p.6).

This study was a part of a wider study involving the design of a novel O/W cream containing active ingredient (X), which has not yet been exploited, for the treatment of an aesthetic condition. The inclusion of the active in the cream was via niosome carriers and, as we explain in the revised paper (p. 8) our method was deemed to be sensitive in detecting the effect of the niosome vesicles on the overall sensorial perception of the creams in terms of pourability, spreadability and firmness.

An extended study including correlation for these products human panel study is necessary to really talk about quantitative sensory interpretation. 

We have added a much clearer explanation in the Introduction (p.2) to link with previous work (see added Reference 19)

The association between rheological measurements and the adhesive ability (tackiness) of pressure-sensitive-adhesives on the skin is well-known [19] and evidences that there is a correlation between user trial data with rheological measurements. The objective of this study was to expand this association to a wider range of sensorial attributes by developing a standard, simple and reliable method for the quantitative assessment of the sensorial attributes of O/W cream formulations by correlating simple sensory lexicons to viscometry (yield stress) and oscillatory (amplitude and frequency sweep) rheological parameters.

Reviewer 2 Report

The manuscript presents interesting aspects, which are well presented.  However, I have some remarks that are presented in the attached file. 

Author Response

Observations to the manuscript Cosmetics-662836 entitled „ Quantitative Sensory
Interpretation of Rheological Parameters of a Cream Formulation".
This paper presents a study on a very important topic in the science of cosmetics: the
sensory evaluation of cosmetic creams and an interesting mode to obtain information
about sensorial properties of cosmetic creams from rheological analyses.
The manuscript presents interesting aspects, which are well presented. However, I
have some remarks:
1. Introduction
– Line 68 – the LVR abbreviation should be explained

We have now defined this abbreviation in the revised paper: Linear Viscoelastic Region.

2. Materials and Methods:
– Lines 85 - 89: the authors should mention the active ingredient and
specify the INCI denominations of ingredients used for the preparation of
these cosmetic creams. Also „triz buffer” is misspelled.

Done.

– Line 92 – reformulation of the first part of the sentence. Also, the authors
should specify what the characteristics of the niosomes were and what
active principle they contained.

Although the preparation and characterisation of the niosomes is not the focus of this paper, we have added more detail on the Methods about the preparation of the niosomes. We have also added a clarification that this study was a part of a wider study involving the design of a novel cream containing active ingredient (X), which has not yet been exploited, for the treatment of an aesthetic condition.

– Table 2 – each stage of usage should be separated by lines to make it
easier to follow

Done

3. Results and discussion:
– Table 4 – the standard deviation values should be added and the table
dimension adjusted to the page dimension.

Done

– Line 184 – the authors mention that the firmness of creams IIIA and IVB is
greater due to the presence of coconut oil. How?

We have revised these sentences to point out the sensitivity of the method in detecting the effect of oils. The effect of coconut oil can be due to its semisolid consistency (compared to the liquid consistency of the other two oils) attributing a self-bodying effect and increased cohesion to the creams.    

– Since the authors analysed creams with and without niosomes it would be
interesting to mention what was the influence of niosomes on sensorial
properties of creams.

We have added an explanation on p. 8: 

"Generally, it was observed that all model creams without niosomes exhibited higher pourability and spreadability scores, with a lower firmness score when compared to their niosome containing counterparts. This shows the sensitivity of the method in detecting the effect of the niosome vesicles on the overall sensorial perception of the creams in terms of pourability, spreadability and firmness."

4. References - the authors should present the all references according to the
recommendations of the Journal.
– Ref. 1 – page range is missing
– The month of publication of articles should not appear in the reference
– Ref. 21 is incomplete

We have now revised and corrected the References.

Round 2

Reviewer 1 Report

No added comments.

Thanks to the authors for their modifications. The manuscrit can be accepted in present form.